# Comparison of Ultrasound Type and Working Parameters on the Reduction of Four Higher Alcohols and the Main Phenolic Compounds

**Qingan Zhang [1,2,*], Hongrong Zheng [2], Shuang Cheng [1], Bowen Xu [2] and Penghui Guo [2]**

1   Henan Key Laboratory of Industrial Microbial Resources and Fermentation Technology, Nanyang Institute of Technology, Nanyang 473000, China; crisycheng@163.com
2   Institute of Food & Physical Field Processing, School of Food Engineering and Nutrition Sciences, Shaanxi Normal University, Xi'an 710062, China; ZHR971011@163.com (H.Z.); 15829905620@163.com (B.X.); 15839819972@163.com (P.G.)
*   Correspondence: qinganzhang@snnu.edu.cn

**Abstract:** In this paper, studies were conducted by a series of single-factor experiments to investigate the effects of ultrasound types and working parameters on the higher alcohols (HA), phenolic compounds, and color properties of red wine, so as to highlight the importance of the comprehensive consideration on its application. The results indicate that ultrasound devices and working parameters do have some definite influences on the HA of wine; moreover, the ultrasound bath (SB-500DTY) is better than the SCIENTZ-950E and the KQ-300VDE. With the SB-500DTY employed to further investigate its effects on phenols and color properties other than on HA, unexpectedly, some variations of color parameters are opposite to the results ever obtained from other ultrasound conditions. In summary, all these results suggest that both the ultrasound type and parameters should be fully considered or neutralized so as to have a comprehensive evaluation about its application, instead of some contradictory results.

**Keywords:** ultrasound; reduction; wine; higher alcohols; phenolic compounds

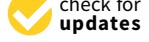



## 1. Introduction

Wine has become a high-end consumer product due to its rich cultural connotation, suitable flavor, and taste; especially for red wine, one of the famous health-promoting alcoholic beverages, there is a great consuming demand and consumer groups all over the world because of its high nutritive and attractive flavor properties [1], which are correspondingly attributed to the rich bioactive compounds occurring in wine, such as phenols, tannins, and flavonoids, and the aromatic ingredients like alcohols, esters, organic acids and ketones. Generally, flavor is one of the important factors assessing wine quality and a key element determining consumer's acceptance [2] and is usually influenced by many factors including yeast strain, fermentation parameters, grape varieties, and maturity [3]. By now, more than one thousand flavor compounds have been identified in wines, such as alcohols, esters, ketones, etc. Although not all these compounds could contribute to the distinctive aroma profiles of wine, the variety of chemical families might give wines a great aromatic complexity, especially together with organic salts/acids, amino acids, sugars, and inorganic acids [4]. Among the flavor-related compounds in wine, higher alcohols (HA) are regarded as one of the most important flavor categories contributing to the organoleptic profile, which can be perceived and widely accepted for its particular importance in food industry [2].

HA are also known as the fusel alcohols, referring to alcohols with a number of carbon atoms from C3 to C8, a very strong and bad-smelling fraction obtained from distilling the fermented products [5]. The boiling points of higher alcohols range from 82.3 °C

(2-propanol) to 225 °C (2-phenylethanol) [6]. Generally, most of the compounds in wine mainly come from three aspects: one is the compounds transferring from grapes, the second is from the yeast metabolism such as alcohols and higher alcohols, and the third is the compounds regeneration during wine aging [7]. As the large group of flavor compounds in wine, HA mainly include the isobutanol, 2-methylpropane-1-alcohol, 2-methylbutane-1-alcohol, 2-phenylethane-1-alcohol, β-phenylethanol, 3-methyl-1-butanol, and n-propanol, etc., which are produced by yeast during the fermentation, either from the grape amino acids (valine, isoleucine, and leucine) via the Ehrlich pathway or directly from sugars [8,9]. For instance, the main aromatic HA (2-phenylethanol) is formed from aromatic amino acid phenylalanine. Generally, most of HA (65%) are derived from the degradation and metabolism of branched-chain amino acids and 35% are derived from the degradation and metabolism of sugars proved by employing the radioactive markers [8,10], and many factors can influence the production of HA including necessary pH, aeration volume, yeast strains, fermentation temperature, variety and maturity of grape berries, etc.

Regarding the impacts of HA on the wine, HA not only influences the flavor but also provides the alcohol moiety needed for the synthesis of desirable esters, consequently changing the organoleptic characteristics of wine [10]. In other cases, it could also mask certain flavors, which is up to the levels of HA. Generally speaking, when the total concentration of higher alcohols in wine is lower than 300 mg/L, higher alcohols have a positive contribution to the aroma form of wine, which can increase the fruit aroma, flower aroma, and aroma complexity of wine [11]. However, at levels above 400 mg/L, a negative effect caused by the apparition of pungent and unpleasant feelings would be noted, which would cause headaches, hangover, and intoxication for the consumers [5]. Furthermore, HA could also be responsible for higher toxicity than ethanol due to their acute toxicity and neurotoxic effects, i.e., significant health concerns [12,13]. Considering the relatively higher concentrations of HA (between 0.4–1.4 g/L) in red wine [14], it is necessary to control the HA content for promoting the quality and popularity of wine.

At present, the approaches available to reduce the HA content in red wine could be classified into two categories: one is by regulating the factors influencing the HA formation during the fermentation process to keep the HA at a lower level, and most studies focused on the modification of microorganism genes [10,15–19]; and the second is to degrade or remove the present HA in fermented red wine by certain physical or chemical methods [14,20–22]. At present, one way to reduce the content of higher alcohols in wine is in the process of red wine brewing and the other is in the finished red wine. In the process of red wine brewing, it is usually necessary to control the initial concentration of amino acids that produce higher alcohols in mash within an appropriate range. The optimization of brewing process and changing different fermentation conditions can generally be used as a way to reduce the content of higher alcohols in red wine; however, it has high production cost, poor effect, and great risk for enterprises; the control methods of higher alcohol content in finished wine mainly include physical aging technology, such as high-voltage pulsed electric field, infrared, electrostatic field, microwave, ultrasonic treatment, etc. [23–27]. Ultrasound, as a new technology with relatively low cost, environmental protection, and non-thermal processing, has received great interest in speeding up the wine ageing in recent years, and it could significantly affect some of the physicochemical and sensory characters of wine due to its mechanical and cavitation effects (ultrasound cavitation can effectively trigger and accelerate some chemical reactions in wine), which resembles the occurrence in natural ageing of wine [25]. In addition, decreasing the content of higher alcohols has also been achieved [14,22]; however, as can be seen in the references, most of the studies were conducted on the different ultrasound baths or probes, and some working parameters of ultrasound are not clearly described regarding its application, which makes it difficult to repeat and compare the results among different groups. Furthermore, the parameters intrinsically related to the ultrasonic equipment (such as the frequency, amplitude of the wave, power, intensity, temperature, and time) have an obvious effect on the outcomes of the technique [28,29]. To the best of our knowledge, no reference was available about the

comparison of ultrasound parameters on reducing the higher alcohols. Therefore, the aim of this research is to compare the effects of different parameters (time, power, frequency, and temperature)/devices (bath and probe) on the HA and some phenolic compounds so as to provide the comprehensive information about the ultrasonic application in winery.

## 2. Materials and Methods

### 2.1. Materials and Equipment

Two batches of wine samples (grape variety of Cabernet Sauvignon: the alcohol content is 12% and sulfur dioxide is added in the fermentation process) were purchased from the Shaanxi Sanxian Winery, Shaanxi Province, China. All the chemicals were of analytical grade in the experiments. The n-propyl alcohol, isobutanol, isopentanol, n-pentanol, and anhydrous ethanol were all of chromatographical purity and bought from Tianjin Komeo Chemical Reagent Co., Ltd., Tianjin, China. The double-distilled water was used throughout the experiment. All the standards were dissolved in methanol to a concentration of 1 mg/mL and stored in darkness at 4 °C before use, which are (+)-catechin (Sigma-Aldrich, Dorset, UK), gallic acid (Alfa Aesar, Shanghai, China), caffeic acid and (−)-epicatechin (National Institutes for Food and Drug Control, Beijing, China), and syringic acid (Shanghai Tauto Biotech Co. Ltd., Shanghai, China), respectively. HPLC-grade methanol was purchased from the Fisher Scientific Co., Ltd. (Marshall Town, IA, USA). Formic acid was bought from the Tianli Chemical Reagent Co. Ltd. (Tianjin, China).

### 2.2. Determinations of HA (N-Propyl Alcohol, Isobutanol, Isopentanol, N-Pentanol) in Red Wine

The contents of HA were separated and determined by gas chromatography of GC-2010PLUS (SHIMADZU Co. Ltd., Shimadzu, Japan) with a column of HP-INNO-WAX (30 m × 0.32 mm i.d., film thickness 0.25 μm, SHIMADZU Co. Ltd., Japan) according to the method by Zhang, et al. [14]. The content of higher alcohols in red wine was determined by gas chromatography. Four peaks could be identified by comparing the retention time of n-propyl alcohol, isobutanol, isopentanol, and n-pentanol with the corresponding standards. The contents of all identified compounds can be calculated from the curve between the standard concentration and the corresponding peak area. The content of higher alcohols is highly correlated with the peak area, and all correlation coefficients are greater than 0.999, indicating that it is feasible to determine the content change of higher alcohols according to the constructed curve. Therefore, this method is used to explore the effect of ultrasonic treatment on the content of higher alcohols in red wine.

The detailed working parameters were as follows: the temperature of the gasification chamber and detector were all set at 210 °C, and the oven temperature was firstly maintained at 30 °C for 1 min and increased to 60 °C at a increase rate of 5 °C/min and held for 1 min, then continuously increased to 100 °C at a rate of 15 °C/min and held for 1 min. Helium was used as the carrier, and its flow rate was of 40 mL/min. A total of 1 μL aliquot of sample (standard solutions or red wine) was injected and the split ratio was of 28:1.

### 2.3. Optimization of Wine Sample Pretreatment for HA Determination

Generally, pretreatment might have a certain effect on the flavor compounds in wine during determination. Considering this, different distillation temperature and time combinations were investigated on a RE-52A rotary evaporator (Zhengzhou Honghua instrument equipment Co., Ltd., (Zhengzhou, China), such as (A) 60 °C, 60 min; (B) 60 °C, 40 min; (C) 60 °C, 30 min; (D) 50 °C, 30 min; (E) 50 °C, 20 min; (F) 40 °C, 30 min; (G) 40 °C, 20 min; and (H) direct injection without distillation. For each sample, 30 mL red wine was evaporated according to the above designated condition in a 100 mL distillation flask, making the final concentrates 10 mL with or without the addition of water.

### 2.4. Sample Preparation by Ultrasound Irradiation

In order to investigate the effect of different ultrasound devices on the content of HA in red wine, the experiments were conducted on the three types of ultrasound instrument,

such as the trough numerical control ultrasound cleaning bath of KQ-300VDE (Kunshan Ultrasonic Instrument Co., Ltd., Kunshan, China), the trough ultrasound multi frequency cleaning bath of SB-500DTY (Ningbo Xinzhi Biotechnology Co., Ltd., Ningbo, China), and the horn ultrasound cell crusher of SCIENTZ-950E (Ningbo Xinzhi Biotechnology Co., Ltd., Ningbo, China).

During ultrasound treatment, 20 mL red wine samples from each group were taken for treatment, the samples were fixed at a specific position of the ultrasound cleaning tank, and the water level of each treatment was checked for consistency; for the horn ultrasound device, the probe was immersed into the sample with 1 cm depth, and the temperature was maintained at $18 \pm 1$ °C with a low-temperature cooling circulating pump. Each treatment was repeated three times.

### 2.4.1. Effects of KQ-300VDE Ultrasound Parameters on the HA of Wine

For the ultrasound of KQ-300VDE, four parameters can be regulated, including ultrasound temperature, time, power, and frequency. With the ultrasound temperature, power, and time fixed at 30 °C, 300 W, and 30 min, the effect was investigated about the ultrasound frequencies on the HA, such as 45, 80, and 100 kHz. Similarly, the exposure times of 10, 20, 30, 40, and 50 min were selected with the other conditions fixed at 30 °C, 45 kHz, and 300 W; the temperature was selected as 25, 30, 35, 40, and 45 °C under the conditions of 300 W, 40 kHz, and 30 min, and the power varied from 180, 210, 240, and 270 to 300 W with the fixed parameters of 30 °C, 45 kHz, and 30 min, respectively.

### 2.4.2. Effects of SB-500DTY Ultrasound Parameters on the HA of Wine

For the ultrasound of SB-500DTY, four variations including the ultrasound frequency, time, temperature, and power were investigated regarding their influences on the HA of wine. For the ultrasound frequency, it varied from 25 and 40 to 59 kHz with the other parameters fixed at 30 °C, 300 W, and 30 min. Similarly, the exposure time changed from 10, 20, 30, and 40 to 50 min with the other parameters kept at 30 °C, 40 kHz, and 300 W; the temperatures were selected as 20, 25, 30, 35, 40, and 45 °C with the other conditions maintained at 300 W, 40 kHz, and 30 min; and the ultrasound power varied from 150, 200, 300, and 400 to 450 W under the conditions of 30 °C, 40 kHz, and 30 min, respectively.

### 2.4.3. Effects of SCIENTZ-950E Ultrasound Parameters on the HA of Wine

For the ultrasound of SCIENTZ-950E, the probe diameter is 6 mm and the frequency varies between 20 and 25 kHz with an automatic tracking mode during working. Thus, only the power variations were investigated from 100, 200, 300, 400, 500, 600, and 700 W with the designated time and temperature of 30 min and $18 \pm 1$ °C, respectively.

### 2.5. Effects of SB-500DTY Ultrasound Irradiation on the Phenols and Color Properties of Wine

In order to further investigate the effect of ultrasound irradiation on the phenols and color of wine other than the HA, the ultrasound of SB-500DTY was employed to treat the red wine sample with the conditions of 30 min, 200 W, 59 kHz, and 30 °C, which was proven to have a good reduction effect on the total HA of wine [14].

### 2.5.1. Determinations of the Phenolic Compounds of Wine

The samples were separated using an Elite HPLC system (Dalian Elite Analytical Instrument Co. Ltd., Dalian, China) equipped with a P230II binary pump, a Rheodyne 20 µL injector loop, and a UV230II Elite detector. Peaks in chromatograms from the EC2006 Elite software were identified and calculated to compare the properties of the corresponding standard and the constructed curves. The column type is TC-C$_{18}$ (5 µm, 4.6 mm × 250 mm, Agilent, Palo Alto, CA, USA). The specific procedure was followed according to [25].

2.5.2. Effects of SB-500DTY Ultrasound Irradiation on the Color Properties of Wine

The treated and untreated samples were scanned by the TU-1810 UV-Visible Spectrophotometer (Beijing Persee General Instrument Co. Ltd., Beijing, China) from 380 to 780 nm with a 1-mm path-length quartz cuvette, and the color characteristics, color density (CD), color hue (CH), and proportions of yellow (%Ye), red (%Rd), and blue (%Bl) were all investigated according to the performance described in the literature [25].

*2.6. Statistical Analysis*

All experiments were performed in triplicate. The results were expressed as the mean ± standard deviations, and the analysis of variance (ANOVA) was employed for the obtained results using the SPSS statistics software of version 11.0 (SPSS Inc., Chicago, IL, USA). The figures were drawn by using the Microsoft Office Excel of 2013 version.

**3. Results and Discussion**

*3.1. Optimization of Wine Sample Pretreatment for HA Determination*

As shown in Figure 1, pretreatment conditions have definite influences on the chromatograms of HA; to be specific, the baseline, peak profile, resolution, and peak numbers, that is to say, different distillation time and water bath temperature, caused a certain loss for HA, the targeted compounds. As a comparison of all the chromatograms in Figure 1A–H, Figure 1H has a better baseline and peak resolution; furthermore, all four individual higher alcohols could be separated and detected, so the samples detected by gas chromatography were selected as the sample without distillation.

*3.2. Effects of Power from Different Ultrasound Devices on the HA Content of Red Wine*

As shown in Figure 2, the content of isopentanol in red wine is the highest among the four higher alcohols. Furthermore, all the ultrasonic powers employed could not only reduce the total HA content of red wine, but also the individual higher alcohol to a certain content. Regarding the ultrasound of KQ-300VDE in Figure 2A, compared with the untreated wine sample, the total HA content of treated samples decreased by 25.83%, 25.63%, 17.40%, 11.48%, and 12.00% with the power of 180, 210, 240, 270, and 300 W, respectively. In the range of less than 210 W, the reduction rate of higher alcohol was faster. When the power was higher than 210 W, the decline rate of higher alcohol slowed down, which may be because cavitation bubbles were easier to form under relatively low ultrasonic power, or the energy released by cavitation bubbles increased when the power was higher. The highest content of isopentanol in red wine decreased quickly with the increase of ultrasonic power from 180 to 210 W, and its reduction rate slowed down with the further increase of power from 240 to 300 W. For the n-pentanol, the higher the power, the lower the reduction rate. The reduction content of isobutanol firstly decreased and then increased with the power increase from 180 to 300 W. Although the content of n-propyl alcohol is lower in wine, the ultrasound power still had a reduction effect on it, and the reduction rate is between 4.6% and 16.3% within the power ranges employed. In a word, the ultrasound power had a definite influence on the HA of wine, and the reduction might be from the degradation caused by the ultrasound cavitation [14].

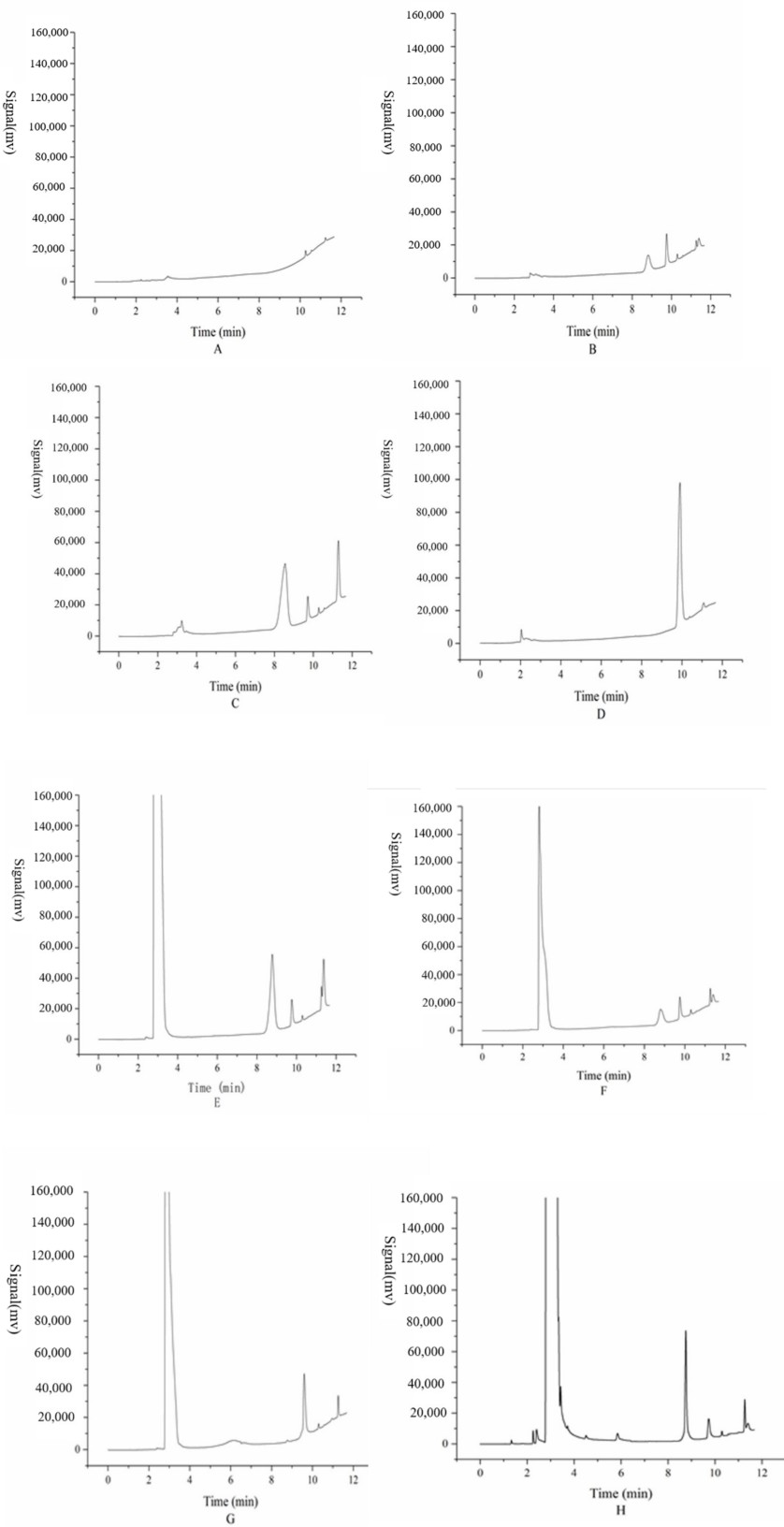

**Figure 1.** Effects of different pretreatments on gas chromatograms of HA in wine.

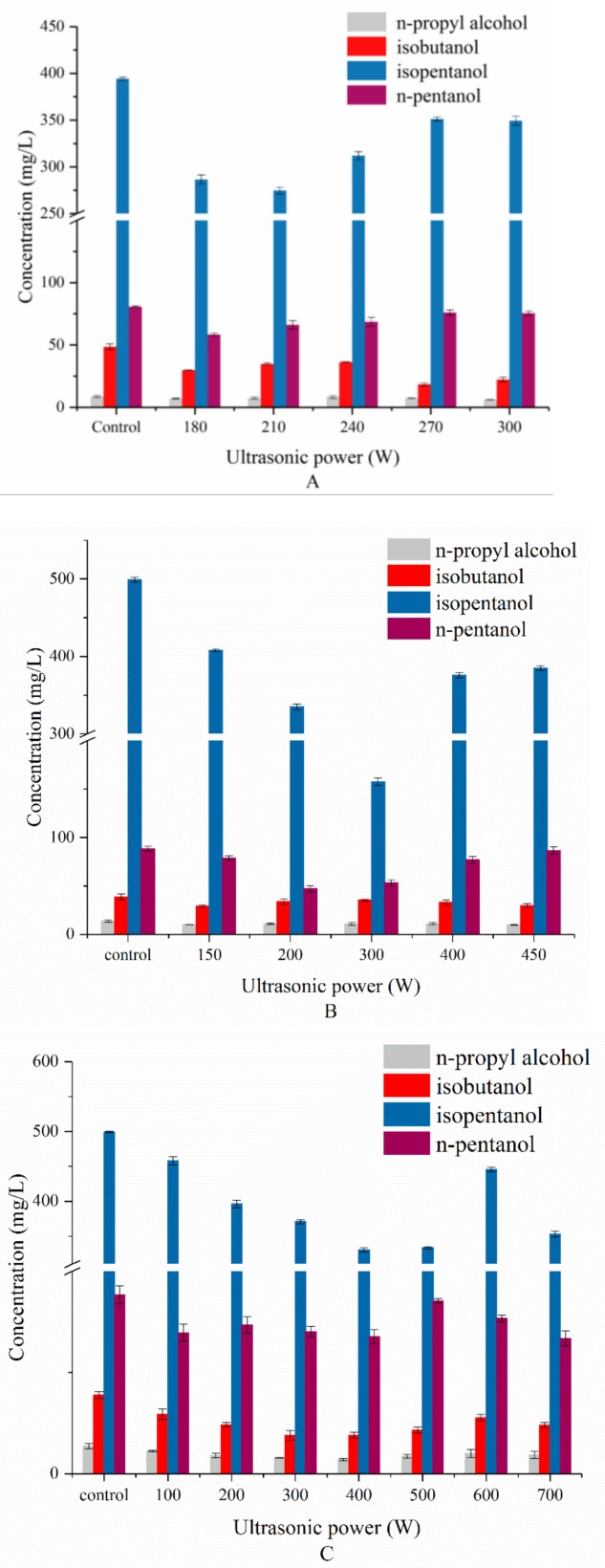

**Figure 2.** Effects of ultrasonic power by KQ-300VDE (**A**), SB-500DTY (**B**), SCIENTZ-950E (**C**) on the HA content of red wine. Note: The ordinate data is too large, so the ordinate is truncated and displayed in the figure.

Figure 2B indicates the effects of SB-500DTY ultrasound power of on the HA content, and it can be seen that this kind of ultrasound bath also had a significant reduction effect on the HA, and the reduction range of total HA was between 25.5% and 39.1% with the power ranging from 150 to 450 W compared with the untreated sample. Furthermore, the SB-500DTY ultrasound had a stronger reduction of total HA than the KQ-300VDE within the powers investigated; for the former ultrasound, the highest reduction was obtained at the power of 300 W, while the latter is the opposite. The reduction of isopentanol and n-pentanol demonstrated a similar trend of first increasing and then decreasing with the increase of power from 150 to 450 W, and the reduction rates reached the maximum of 68.47% and 46.45% at 300 and 200 W, respectively. In comparison, the degradation of isobutanol and n-propyl alcohol was relatively lower under the ultrasound powers employed, and their highest reduction rates of 25.4% and 26.7% were obtained for the powers of 150 and 450 W, respectively.

It can be seen from Figure 2C, the reduction rates of the total HA content were between 11.27% and 33.80% within the powers of 100 to 700 W, and the highest and lowest reduction rates were obtained at the power of 400 and 100 W, respectively. In general, the total HA content decreased with the power increased from 100 to 400 W; then, it increased from 500 to 600 W and decreased at 700 W. Additionally, this trend might be attributed to the variations of isopentanol and n-pentanol, since their contents are higher than the others. It can be seen in Figure 2C that the price reduction rate of the four higher alcohols at 600 W decreased and then increased compared with the lower power, which may be due to the fact that higher ultrasonic power will produce higher heat and accelerate the reaction between aldehydes and acids in red wine, while the newly generated higher alcohols have no time to degrade and then have a short increase. Considering that too-high ultrasound treatment power will destroy the sensory quality and nutritional value of red wine, too-high power should not be used as the research condition in the selection of ultrasonic treatment power.

Overall, it can be seen that different ultrasound devices do have some differences in reducing the HA of wine, even at the same employed power, which might be attributed to the different efficiency converted from electric power to sonochemistry power of the different instruments [29]. As a comparison, the SB-500DTY ultrasound had a better reduction efficiency on the HA of wine under the used powers. Generally, the reduction content of HA would increase with the rising of ultrasound power, but they might not be positively correlated with each other, as shown in Figure 2, which is also in accordance with the results by the authors of Zhang et al. [30]. Regarding the effect of ultrasound power, through the comparison of three ultrasound treatment methods, it can be found that, under the horn ultrasound treatment method, high power cannot have a greater degradation effect on higher alcohols, and there was a certain limit on the degradation of higher alcohols; when the two tank ultrasound treatment methods were used, it can be found that the multi-frequency ultrasound treatment power had a more significant effect on the degradation rate of higher alcohols.

### 3.3. Effects of Frequency from Different Ultrasound Devices on the HA Content of Red Wine

As shown in Figure 3, all the frequencies employed had a definite reduction effect on the HA content, either the ultrasound of KQ-300VDE or SB-500DTY. In the meantime, the contents of the four individual higher alcohols in red wine all decreased under the treatment of different ultrasound frequencies in comparison with the untreated wine samples. In Figure 3A, the total reductions of HA were 35.91%, 39.46%, and 30.85% with the corresponding frequencies of 45, 80, and 100 kHz, respectively. To be specific, the degradation rate of isopentanol was the most obvious, and reached the maximum of 36.68% at the frequency of 45 kHz, and it was lower at the frequency of 80 and 100 kHz. However, for the n-pentanol and isobutanol, they had a similar reduction trend, with the highest reduction obtained at the frequency of 80 kHz, and the others were lower. The content of n-propyl alcohol was lower, but its degradation rate was more similar to the other three

higher alcohols, which were 27.86%, 34.00%, and 34.35% at the corresponding frequencies of 45, 80, and 100 kHz, respectively.

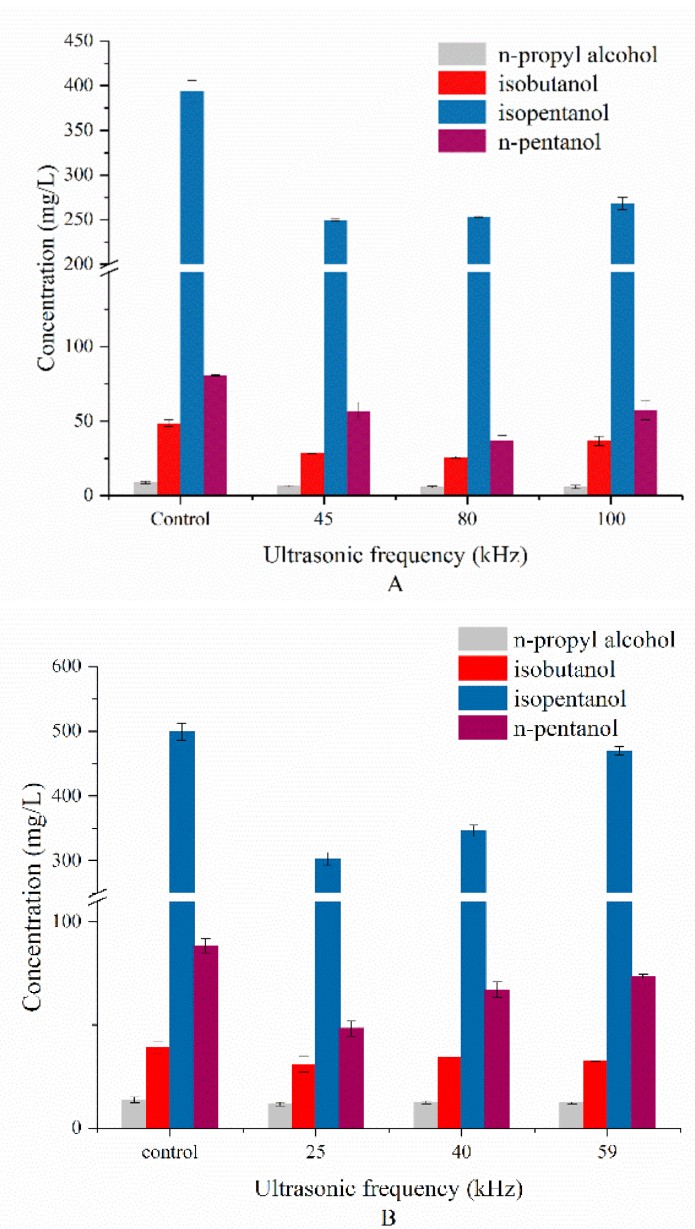

**Figure 3.** Effects of ultrasonic frequency by KQ-300VDE (**A**), SB-500DTY (**B**) on the HA content of red wine. Note: The ordinate data is too large, so the ordinate is truncated and displayed in the figure.

As shown in Figure 3B, both isopentanol and n-pentanol had a similar decrease trend upon the employed frequencies from 25 to 59 kHz; moreover, the higher the frequency, the less obvious the HA reduction. Both of them obtained the highest reduction rate of 39.32% and 45.42% at the frequency of 25 kHz, and the lowest decrease rate of 5.91% and 16.77% at the frequency of 59 kHz, respectively. For the n-propyl alcohol and isobutanol, they also had a similar changing trend, and their highest reduction rates of 14.97% and 20.44% were obtained at the frequency of 25 kHz, while the lowest reduction rates of 9.18% and 12.37% were obtained at the frequency of 40 kHz. As can be seen from Figure 3, the ultrasound of SB-500DTY had a better reduction effect on the HA than that of KQ-300VDE.

Regarding the effect of frequency on the HA of wine, it could be explained by these facts and phenomena; the frequency can influence the acoustic cavitation and therefore,

two main influences: the mechanical breaking from the collapsing bubbles of acoustic cavitation in the water solution near the surface and the generation of free radicals from the breakdown of the water inside the bubbles [28]. Furthermore, the ultrasonically mechanical effects are not very noticeable at the higher frequencies, due to the insufficient negative pressure to generate the acoustic cavitation during the period of the rarefaction or the insufficient time for the bubbles to collapse during the compression [31]. In addition, the chemical effects from the cavitation are also influenced by the ultrasound frequency, as Ashokkumar [32] demonstrated how the production of the free radicals increased at the higher frequencies. Generally, at the lower frequencies, high disruptive forces may be expected because of the violently collapsing bubbles from the locally instant high temperatures and pressures.

### 3.4. Effects of Exposure Time from Different Ultrasound Devices on the HA Content of Red Wine

As shown in Figure 4, the HA of wine treated by ultrasound irradiation were significantly reduced in comparison with the untreated wine. Overall, the reduction rates demonstrated an increase trend with the increase of ultrasound exposure time. As can be seen from Figure 4A, the total HA in wine decreased from 532.39 to 411.96 mg/L with the increase of exposure time from the beginning to 40 min, while it was 424.36 mg/L at 50 min, i.e., the reduction rate decreased with the further extension of time. For the individual higher alcohol, the isobutanol and n-propyl alcohol content in wine varied with a similar trend, it decreased with the time increase from 10 to 30 min, that is to say, the highest reductions regarding the isobutanol and n-propyl alcohol was all obtained at the exposure time of 30 min. Additionally, for the isopentanol, its highest reduction was obtained at the ultrasound time of 40 min; for the n-pentanol, its highest reduction was obtained at the ultrasound time of 10 min.

In Figure 4B, the reduction of n-pentanol decreased with the increase of exposure time from 10 min to 50 min, while for n-propyl alcohol and isobutanol, the reduction rate decreased with the extension of ultrasound time from 10 to 40 min, and the highest reductions were of 15.53% and 22.46% at 10 min for n-propyl alcohol and isobutanol, respectively. The reduction of isopentanol reached the maximum of 34.07% at the ultrasound time of 40 min. Overall, the total reduction of HA in wine presented a whole increasing trend with the extension of exposure time from 10 to 40 min, which is in accordance with the isopentanol. As a comparison, the KQ-300VDE ultrasound had a maximum reduction of the total HA of 22.62% at 40 min, while for the SB-500DTY ultrasound, it was 29.41% at 40 min, that is say, the latter had a better reduction of total HA than the former; therefore, the effect of the SB-500DTY ultrasound is better than the KQ-300VDE ultrasound in reducing higher alcohols.

As regards to the variation trend of the total HA influenced by ultrasound exposure time, it might be related to the generation of free radicals and their consequent propagated chain reactions. In the beginning, the ultrasound cavitation triggers some free radicals; at this stage, the free radicals are generated and accumulated and simultaneously oxidize the compounds in wine such as HA, so the HA are not significantly decreased. Additionally, with the extension of ultrasound time, some reactions initiated by free radicals become violent, and more targeted molecules will be attacked due to the free radicals extremely exceeding the threshold of reaction between HA and free radicals, which would result in a quick reduction of HA.

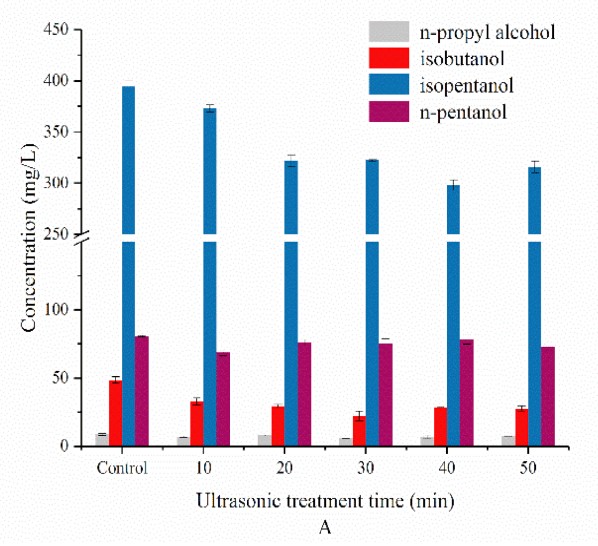

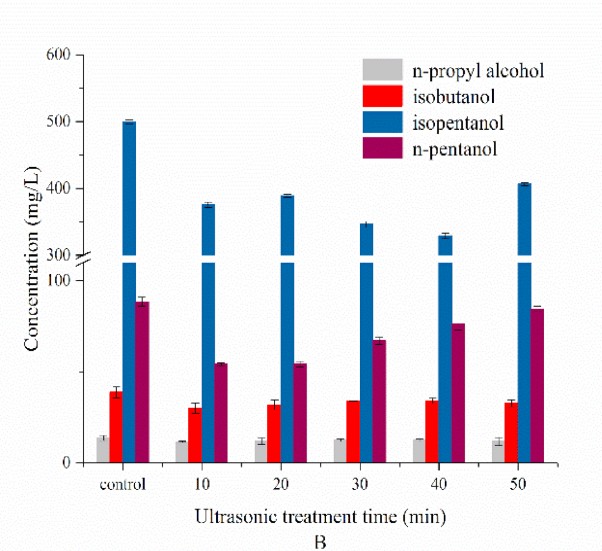

**Figure 4.** Effects of ultrasonic time by KQ-300VDE (**A**), SB-500DTY (**B**) on the HA content of red wine. Note: The ordinate data is too large, so the ordinate is truncated and displayed in the figure.

### 3.5. Effects of Temperature from Different Ultrasound Devices on the HA Content of Red Wine

Figure 5 demonstrates that all the ultrasound temperatures could reduce the contents of total HA and the four individual higher alcohols in red wine to a certain extent. In Figure 5A, compared with the HA content in untreated wine, the total HA contents decreased by 22.25%, 20.12%, 29.49%, 31.98%, and 30.99% at the corresponding temperatures of 25, 30, 35, 40, and 45 °C, respectively. Additionally, the whole changing trend of the total HA is consistent with that of the isopentanol and n-pentanol, i.e., the reduction of the total HA, isopentanol, and n-pentanol demonstrated a whole increasing trend within the studied temperature, due to the fact that the isopentanol and n-pentanol are the main contributors of the total HA. For the individual higher alcohols of n-propyl alcohol and isobutanol, their highest reductions of 42.31% and 26.27% were obtained at the temperature of 25 and 40 °C, respectively.

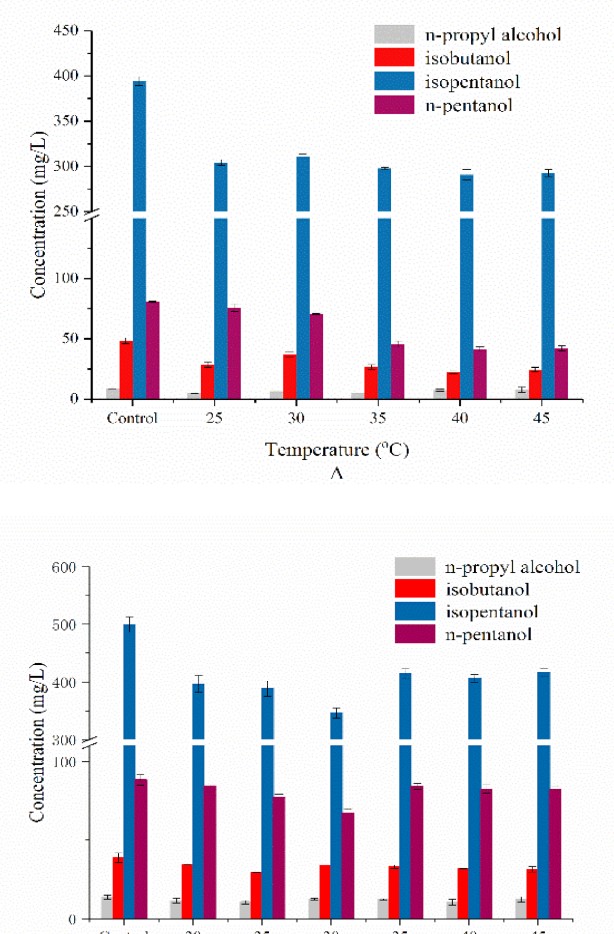

**Figure 5.** Effects of ultrasonic temperature by KQ-300VDE (**A**), SB-500DTY (**B**) on the HA content of red wine. Note: The ordinate data is too large, so the ordinate is truncated and displayed in the figure.

As shown in Figure 5B, the total HA content demonstrated a trend of first decreasing and then increasing with the increase of temperature from 20 to 45 °C, and the highest reduction was obtained at the temperature of 30 °C. The individual higher alcohols of isopentanol and n-pentanol have the similar changing trend of the total HA. For the n-propyl alcohol and isobutanol, the highest reductions of 24.60% and 23.87% were obtained at the ultrasonic temperature of 25 °C. As a comparison, the ultrasound temperature had a similar reduction effect on the HA either for the ultrasound of KQ-300VDE or SB-500DTY.

In two different trough ultrasound treatment methods, the content of higher alcohols in red wine decreased slowly with the increase of ultrasound temperature. It may be due to the cavitation effect caused by ultrasonic treatment and the high temperature in the small space, which accelerated the oxidation of alcohols and the esterification reaction between substances. Generally, the impact of ultrasound temperature on the molecules stability in liquid is a rather complicated phenomenon, because the temperature can influence the surface tension and gas solubility and the solutes' vapor pressure. Temperature increase will cause the surface tension to decrease, resulting in reducing the intensity of threshold to produce the acoustic cavitation. Furthermore, the collapsing temperature would also decrease with the increase of liquid temperature, due to the decreasing of the viscosity and/or surface tension. Therefore, liquid temperature increase would result in a lower degradation of HA, and the variations of reduction rates at different temperatures might be attributed to the combined cavitation and thermal effects. To be specific, from the acoustic cavitation viewpoint, temperature increasing has a negative influence on the reduction due to the decreased intensity of cavitation, while from the thermal effect,

the increase of temperature presented a positive effect due to the increased oxidation reaction, so an optimal temperature might exist upon the performance of ultrasound [30,31].

### 3.6. Effects of Treatment by SB-500DTY Ultrasound on the Phenols and Color Properties of Red Wine

As mentioned above, both ultrasound types and ultrasound parameters could affect the reduction rates of HA, reducing the content of higher alcohols that affected the sensory quality of red wine and improving the sensory quality of red wine. Furthermore, the SB-500DTY ultrasound had a relative higher efficiency than that of the KQ-300VDE. Therefore, the former was employed to further investigate its effect on the phenols and color properties of wine other than the HA, since they are also the important factors determining the wine quality and consumer's selection.

### 3.6.1. Effects of Treatment by SB-500DTY Ultrasound on the Phenols of Red Wine

As shown in Figure 6, the peaks of red wine without ultrasound treatment and ultrasound treatment were compared with corresponding standards by HPLC, five phenolic compounds were separated and identified, and the contents were calculated on the constructed curves with standards, and all the correlation coefficients were above 0.999, indicating the feasibility of the determination conditions.

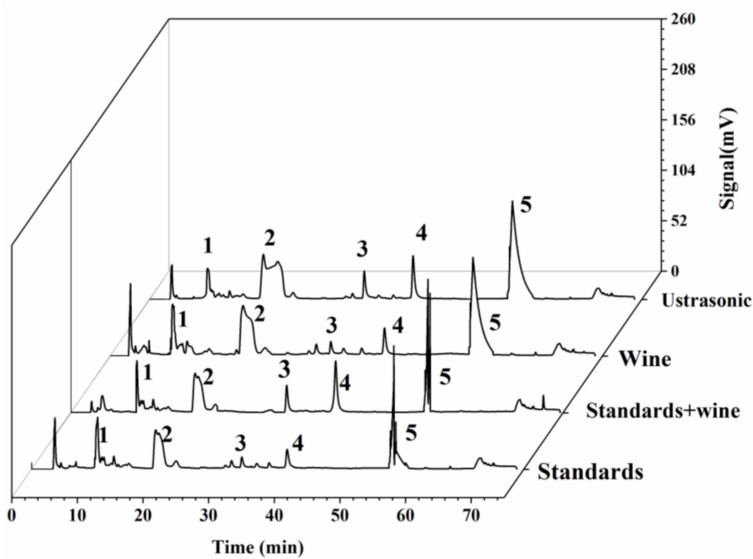

**Figure 6.** Chromatograms of red wine samples obtained by HPLC at 280 nm. (1) Gallic acid; (2) (+)-Catechin; (3) Caffeic acid; (4) Syringic acid; (5) (+)-Epicatechi.

Anthocyanins mainly contribute to the wine color of red or purple, and their content and compositions influence the color property of red wine. Figure 7 indicated that all the contents of the identified phenolic compounds except the gallic acid in red wine had a decrease after ultrasonic treatment, especially for the (+)-catechin, syringic acid, and (+)-epicatechin; they were significantly reduced after ultrasound treatment. For the catechin in wine, it can polymerize with the anthocyanins to make the pigments more stable than the free anthocyanins during wine aging. Considering the possible effect of ultrasound on assisting the wine aging process, the decrease of catechin content might be attributed to the auxiliary color of anthocyanins [25]. Regarding the (+)-epicatechin, it is a monomer, and its polymer is procyanidins, and its reduction may be due to its continuous participation in the cleavage and polymerization of procyanidins polymer or its interconversion with (+)-catechins. The decrease of syringic acid might suggest that the self-degradation of the malvidin-3-O-glucoside was inhibited at the employed ultrasound conditions, since it is one of the main degraded compounds, which is inconsistent with the results obtained from the other ultrasonic conditions, or ultrasound promoted some reactions between syringic

acid and other compounds. The increase of the gallic acid remains unknown, while it is in accordance with the results in the literature [25]. In summary, the ultrasound conditions utilized for reducing the HA did have some influence on the phenolic compounds of wine, and these changes brought by ultrasound treatment may help to accelerate the aging of wine.

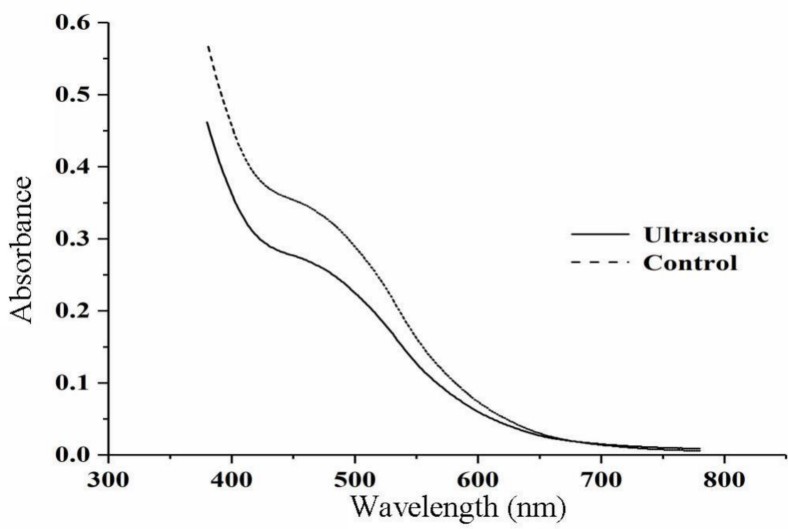

**Figure 7.** Effects of ultrasound treatment on the contents of gallic acid, (+)-catechin, caffeic acid, syringic acid, (+)-epicatechin in wine samples.

### 3.6.2. Effects of Treatment by SB-500DTY Ultrasound on the Color Properties of Red Wine

Generally, phenols in red wine are related to the wine color, and their variations may cause the changes of color index. As shown in Figure 8, the absorption spectrum curve of red wine after ultrasound treatment was similar to that of red wine without ultrasound treatment, and its absorbance decreased after ultrasound treatment, indicating that ultrasound treatment had a significant impact on the color presentation of red wine. Regarding the specific color parameters, it can be seen in Table 1 that some of them were significantly decreased in the ultrasonically treated wine, such as wine color ($A_{520}$), color density (CD, $A_{420} + A_{520} + A_{620}$), browning index ($A_{420}$), percentage of red color (%, $A_{520}/CD$), and b * value, while the percentage of blue color (%, $A_{620}/CD$), L * values, and a * values increased significantly. Overall, these variations of the color parameters are opposite to the results using ultrasound accelerating the wine ageing in literature [25], which might be due to the different ultrasound devices and working conditions and wine variety.

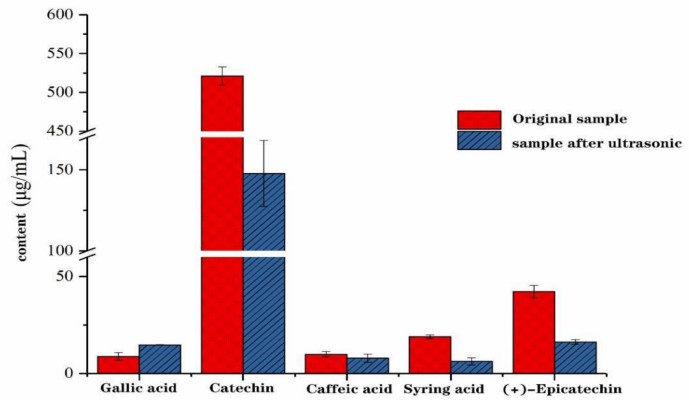

**Figure 8.** Effects of ultrasonic treatment on the visible spectra of wine samples. Note: The ordinate data is too large, so the ordinate is truncated and displayed in the figure.

**Table 1.** Effects of ultrasound treatment on the wine color properties.

| Color Parameters | Original | Ultrasonic Treatment |
|---|---|---|
| Wine color | 0.24 ± 0.00 [a] | 0.19 ± 0.00 [b] |
| Color density | 0.68 ± 0.01 [a] | 0.54 ± 0.00 [b] |
| Browning index | 0.38 ± 0.01 [a] | 0.30 ± 0.00 [b] |
| Tone value | 1.57 ± 0.02 [a] | 1.60 ± 0.00 [a] |
| Percentage of red color (%) | 61.44 ± 0.09 [a] | 56.53 ± 0.00 [b] |
| Percentage of yellow color (%) | 35.91 ± 0.00 [a] | 35.29 ± 0.00 [a] |
| Percentage of blue color (%) | 7.78 ± 0.00 [a] | 8.17 ± 0.00 [b] |
| $L^*$ | 52.36 ± 0.09 [a] | 52.81 ± 0.06 [a] |
| $a^*$ | 40.82 ± 0.24 [a] | 40.76 ± 0.14 [a] |
| $b^*$ | 30.67 ± 0.64 [a] | 27.84 ± 0.36 [b] |

Different lowercase letters within the same row represent the significant differences between the original wine and ultrasonic wine. Note: different letters (a, b) in the same line in the table indicate significant differences (*: $p < 0.05$).

## 4. Conclusions

In summary, the results indicate that different ultrasound devices and working parameters (temperature, power, time, and frequency) do have some definite influence on the HA of wine; as a comparison, the ultrasound of SB-500DTY has a better reducing effect on HA than that of the other types (KQ-300VDE, SCIENTZ-950E). Furthermore, the cleaning ultrasound bath (KQ-300VDE, SB-500DTY) is better than the probe type of ultrasound (SCIENTZ-950E). In order to further investigate the effects of this kind of ultrasound on the phenols and color properties of wine other than on the HA, the optimized ultrasound conditions of SB-500DTY, at which there had a maximal reduction of HA, were employed to treat the wine. Unexpectedly, the variations of color parameters are opposite to the results ever obtained from other ultrasound conditions, which proved to have a positive accelerating effect on the wine ageing. In other words, all the results suggest that not only the ultrasound type but also the working parameters should be fully considered or neutralized so as to have a comprehensive evaluation about its application, instead of the contradictory results, such as the reduction of HA and color parameters, for which the reduction of HA is what we need, but the decrease of the latter is not.

**Author Contributions:** Providing experimental funding, designing experiment, experiment devices, and revising and editing the manuscript, Q.Z.; writing the original draft, drawing the figures, and editing manuscript, H.Z.; collecting background information, designing the experiment, and providing experimental funding, S.C.; designing the experiment, carrying out experiments, and the statistical analysis, B.X.; writing the original draft, P.G. All authors have read and agreed to the published version of the manuscript.

**Funding:** This study was funded by National Natural Science Foundation of China [grant number 31972206], the Innovation Talents of Science and Technology Serving Enterprise Project of Xi'an, Shaanxi Province, China [grant number 2020KJRC0011], the Fundamental Research Funds for the Central Universities of China [grant number GK202102009], and Henan Key Laboratory of Industrial Microbial Resources and Fermentation Technology, Nanyang Institute of Technology, Henan Province, China [grant number HIMFT20190306].

**Conflicts of Interest:** The authors declare that they have no known competing financial interests or personal relationships that could have appeared to influence the work reported in this paper.

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
