# Peer review of "Comparison of Ultrasound Type and Working Parameters on the Reduction of Four Higher Alcohols and the Main Phenolic Compounds"

_sustainability, doi:10.3390/su14010417_

Round 1

Reviewer 1 Report

I just have to congratulate the authors on such a fine manuscript and very interesting results and presentation. Looking forward to the next research.

Author Response

Thank you very much for the reviewer’s comments. We will continue to work hard.

Reviewer 2 Report

There are lots of place for improvement of an article. I mention those I think are the important ones:

  • the authors did not use a template file during the preparation of the article. This has made reviewing process very hard
  • the authors used several very old textbooks and handbooks, over 15 years old (one from 1986.!!!). I suppose those informations are questionable.
  • there is no data about "naturally" aged wines for comparison with "artificially" made ones. These results can be observed and compared for not properly kept and made wines
  • why influence of pH did not consider? Changing pH red wine can be transformed to white and vice-versa.
  • the premise of scientific work did not explain properly    

Author Response

We appreciate for the reviewers’ warm work earnestly, and hope that the correction will meet with approval.

Once again, thank you very much for your careful work, helpful comments and professional suggestions.

Reviewer 3 Report

This is an interesting and well performed experimental work on the accelerated ageing of red wine components by use of different types of ultrasound devices.

The authors should address the following issues to further improve the quality of their presentation:

Line 96: More details on the composition of the wine samples should be included: How much is the alcohol content? Which additives/ color agents etc have been added in the fermentation production process?

Line 148 For the ultrasound KQ-300VDE, maximum power is 300 W. Some way of normalization should be made in order to allow a fair comparison with the other two types.

Line 158 For the ultrasound SB-500DTY, maximum power is 450 W. Normalization is necessary in order to allow a fair comparison with the other two types.

Line 168 For the ultrasound SCIENTZ-950E, maximum power is 700 W. The authors should normalize somehow in order to allow a fair comparison with the other two types.

Line 174 why only the effects of ultrasound SB-500DTY are tested on the phenols and color properties of the wine samples? What about the effect of the other two types of ultrasound?

Lines 216-217: increasing power initially leads to increased reduction on the total HA content, however this trend is reversed after 210 W. This needs further discussion. Line 217 you forgot to add the 270 W.

Figure 2C the behavior at 600 W power indicates a singularity (or statistical error?). Please discuss.

Figures are not very well readable: fonts must increase in size and bar graphs enhanced.

Author Response

(The authors gave the same response as above.)

Round 2

Reviewer 2 Report

The authors did not put too much effort in revising an article. The revising process is in fact adding three sentences and an one reference. Answers on my comments are not concise. Due the mentioned facts, I cannot propose accepting the article in this form. 

Author Response

We sincerely thank the reviewers for their enthusiastic work and hope that the correction will be approved. See the word document for specific answers and modifications.

Reviewer 3 Report

The paper may be published in the updated version

Author Response

We sincerely thank the reviewers for their enthusiasm and careful work. We will continue to work hard.

Round 3

Reviewer 2 Report

I suggest for accepting this paper